# Post-Fracture Inpatient and Outpatient Physical/Occupational Therapy and Its Association with Survival among Adults with Cerebral Palsy

**DOI:** 10.3390/jcm11195561

**Published:** 2022-09-22

**Authors:** Daniel G. Whitney, Tao Xu, Daniel Whibley, Dayna Ryan, Michelle S. Caird, Edward A. Hurvitz, Heidi Haapala

**Affiliations:** 1Department of Physical Medicine and Rehabilitation, University of Michigan, Ann Arbor, MI 48109, USA; 2Institute for Healthcare Policy and Innovation, University of Michigan, Ann Arbor, MI 48109, USA; 3Kidney Epidemiology and Cost Center, School of Public Health, University of Michigan, Ann Arbor, MI 48109, USA; 4Department of Orthopaedic Surgery, University of Michigan, Ann Arbor, MI 48109, USA

**Keywords:** cerebral palsy, physical therapy, occupational therapy, fracture, bone fragility, mortality

## Abstract

Physical and/or occupational therapy (PT/OT) may improve post-fracture health and survival among adults with cerebral palsy (CP), but this has not been studied in the inpatient setting. The objective was to quantify the association between acute inpatient and outpatient PT/OT use with 1-year mortality among adults with CP. This was a retrospective cohort study of adults with CP with an incident fragility fracture admitted to an acute care or rehabilitation facility using a random 20% Medicare fee-for-service dataset. Acute care/rehabilitation PT/OT was measured as the average PT/OT cost/day for the length of stay (LOS). Weekly exposure to outpatient PT/OT was examined up to 6 months post-fracture. Cox regression examined the adjusted association between the interaction of acute care/rehabilitation average PT/OT cost/day and LOS with 1-year mortality. A separate Cox model added time-varying outpatient PT/OT. Of 649 adults with CP, average PT/OT cost/day was associated with lower mortality rate for LOS < 17 days (HR range = 0.78–0.93), and increased mortality rate for LOS > 27 days (HR ≥ 1.08) (all, *p* < 0.05). After acute care/rehabilitation, 44.5% initiated outpatient PT/OT, which was associated with lower mortality rate (HR = 0.52; 95% CI = 0.27–1.01). Post-fracture inpatient and outpatient PT/OT were associated with improved 1-year survival among adults with CP admitted to acute care/rehabilitation facilities.

## 1. Introduction

Fractures represent a quadruple threat to adults with cerebral palsy (CP) as they are common, begin to accumulate early in life [1], associated with premature morbidity and mortality [2,3,4,5], and are economically costly [6]. Clinical rehabilitation, including physical and/or occupational therapy (PT/OT), may be effective at mitigating post-fracture health declines, but this has been seldom studied among adults with CP.

Within the U.S. healthcare system, there are different post-fracture rehabilitation pathways that may improve health outcomes in the general population, particularly if rehabilitation is initiated early [7,8,9,10]. We recently characterized post-fracture rehabilitation pathways among a large cohort of adults with CP with an incident fragility fracture to better understand the current fracture care landscape. The majority (>70%) were discharged home without acute care or inpatient rehabilitation, and more than half of this home discharge group did not initiate outpatient PT/OT within 6 months post-fracture, potentially impacting recovery [11]. For example, this study found that PT/OT use within 6 months post-fracture among those with a home discharge (without acute care/rehabilitation) was associated with improved 1-year survival [11]. While the mechanisms were not examined, these findings suggest that outpatient PT/OT may improve post-fracture health among adults with CP.

The effectiveness of early rehabilitation applied in acute care and inpatient settings to mitigate post-fracture health declines has not been investigated for adults with CP. In our recent study [11], ~1 in 5 adults with CP with a fragility fracture were admitted to acute care/rehabilitation facilities, but the effect of post-fracture rehabilitation on outcomes was not examined in this cohort (only in the home discharge cohort). The goal of these facilities is to restore enough function through early and/or intensive rehabilitation, predominately PT/OT, in a short time span and address adaptive equipment needs (e.g., assistive mobility devices) to enable safe home discharge of the patient as quickly as possible. The design and implementation of acute care/rehabilitation PT/OT depends in part on several patient-level factors, such as issues with fracture healing, pain, and the ability to perform and tolerate therapies. Adults with CP have long-standing issues with pain, function, and medical complexity, which may be exacerbated by a fracture [12,13,14,15,16,17,18]. These factors may alter the initiation and volume of PT/OT sessions and the duration (i.e., number of days) that inpatient PT/OT services are applied in the acute care/rehabilitation setting. Taken together, there may be a spectrum of acute care/rehabilitation PT/OT use patterns that may differently associate with post-fracture outcomes.

A better understanding of how acute care/rehabilitation PT/OT use associates with post-fracture outcomes among adults with CP admitted to acute care/rehabilitation facilities could inform post-fracture care. Further, determining if outpatient PT/OT after acute care/rehabilitation improves post-fracture outcomes may identify additional opportunities to improve post-fracture care for this population. Accordingly, the primary objective of this study was to determine the association between acute care/rehabilitation PT/OT use with 1-year mortality among adults with CP that sustained an incident fragility fracture. We hypothesized that higher acute care/rehabilitation PT/OT volume would be associated with improved 1-year survival conditional on shorter acute care/rehabilitation lengths of stay (i.e., interaction between PT/OT volume and length of stay). The secondary objective was to determine if initiating outpatient PT/OT within 6 months post-fracture was associated with 1-year survival. We hypothesized that initiating outpatient PT/OT within 6 months post-fracture would be associated with improved 1-year survival even after accounting for acute care/rehabilitation PT/OT.

## 2. Materials and Methods

### 2.1. Data Source

This retrospective cohort study used claims data from a random 20% sample of the Medicare fee-for-service database (Centers for Medicare & Medicaid Services, Baltimore, MD, USA) Part A (hospital insurance) and Part B (medical insurance. This study did not have access to Part C (Medicare Advantage Plan) or Part D (medication prescription coverage). Medicare is a federal program in the U.S. providing health insurance to adults ≥65 years of age, individuals with certain disabilities including CP at any age, or individuals with end-stage renal disease. Individuals that are enrolled in Medicare can have dual enrollment with Medicaid. Medicare pays first for Medicare-covered services that are also covered by Medicaid, including the diagnostic and rehabilitation-based services needed to conduct this study without bias from missing information from those with dual enrollment in Medicare and Medicaid.

Claims data are primarily used for billing reimbursement of healthcare services, but can be used for research purposes to identify medical conditions and healthcare service use based on unique codes attached to claims. The codes used to identify the variables in this study are presented in Appendix A. The data for this study are de-identified prior to the researcher access. Therefore, patient consent was not required and the University’s Institutional Review Board approved this study as non-regulated (HUM00158800).

### 2.2. Cohort Selection

A flow chart to derive the final sample is presented in Appendix A. This study included adults ≥18 years with CP that had an incident fragility fracture (index data = fracture date) at an identifiable site between 1 January 2014–31 December 2017 with an admission to an acute care and/or an Inpatient Rehabilitation Facility. This study combined acute care and acute inpatient rehabilitation services because PT/OT is often applied early in the acute care setting, which can inform admission decisions to an Inpatient Rehabilitation Facility. Thus, combining services provides a more comprehensive assessment of the extent and timing of acute PT/OT.

To be included in the analysis, adults with ≥12 months of continuous enrollment in Part A and B prior to the fracture date (for baseline data) and with ≥30 days of continuous enrollment in Part A and B after the fracture date were included. This study excluded adults that died within 30 days post-fracture as they may represent excessively frail individuals that may not benefit from post-fracture rehabilitation [19]. This study excluded adults that were admitted to a Skilled Nursing Facility because these patients and the rehabilitation treatment are fundamentally different. Finally, adults with a combined acute care/rehabilitation length of stay >31 days were excluded (*n* = 22, ~3%) as this length of stay is atypical and may represent more complex cases presenting with a constellation of complications unrelated to the fracture.

Adults with CP were identified by ≥1 inpatient or ≥2 outpatient claims (with a pertinent code for CP) on separate days within 12 months of one another. A fragility fracture was identified as a ≥1 inpatient or outpatient claim (with a pertinent code for fracture) without a trauma code (e.g., car accident) 7 days before to 7 days after the index fracture date [2,20].

### 2.3. Acute Care/Rehabilitation PT/OT

This study had access to the Medicare Provider Analysis and Review (MEDPAR) file, which aggregates entire inpatient stays into a single claim. The single claim provides a length of stay and total charges for PT and OT services provided during the acute care/rehabilitation stay, but it does not provide dates of PT or OT services, number of PT or OT sessions, or any details of the PT or OT sessions (e.g., activities). We developed a “volume” measure of inpatient PT/OT as the average cost/day: after standardizing the PT and OT costs to 2017 U.S. dollars, the standardized cost was divided by the acute care/rehabilitation length of stay for each person, e.g., USD 10,000 for PT/OT services during a 10-day stay in the acute care/rehabilitation setting = USD 1000/day. The assumption is that higher average cost/day reflects a greater volume of therapy during the inpatient stay. This study examined PT/OT and not PT and OT separately to capture physical-based rehabilitation that is more inclusive of the varied functional abilities of the population with CP, and to be consistent with how outpatient PT/OT was captured in this study, as described below.

To assess if the average PT/OT cost/day measure was driven by geographical differences in therapy service costs, we developed a generalized linear model with gamma distribution and log-link function, which is appropriate for analyzing healthcare cost data [21,22,23]. This model estimated the marginal means and cost ratio of average PT/OT cost/day for two geographical variables (in separate models) after adjusting for age, gender, race, dual eligibility with Medicaid, comorbidities, acute care/rehabilitation length of stay, and fracture site. The results of the models suggest that the average PT/OT cost/day did not differ across the 4 U.S. regions or 9 U.S. Divisions (Appendix A) (findings were similar when PT and OT cost/day were examined separately).

### 2.4. Outpatient PT/OT

“Initiation” of outpatient PT/OT is defined in this study as attending the first PT/OT session post-fracture, as opposed to scheduling a PT/OT appointment. Outpatient PT/OT was identified from outpatient files (i.e., Outpatient, Part B, Home Health Agency) using a comprehensive list of codes for physical, occupational, and other therapies of a physical nature (excludes PT/OT evaluations) [24]. Some of these codes cannot distinguish between therapy types or can be used interchangeably for outpatient and home settings. To avoid misclassification, codes were combined into a single indicator of any outpatient PT/OT use.

While the MedPar file (to capture inpatient PT/OT) aggregates all services into a single claim for the entire inpatient stay, outpatient files contain individual claims for each service provided based on the date of service, allowing for a more granular assessment of the date of services. For the post-fracture period, dichotomous variables were created to indicate outpatient PT/OT use per week (starting at week 1) through 6 months post-fracture based on the date of the service claim. A 6-month period was selected to capture an early post-fracture period that may be crucial to implement therapies to improve long-term outcomes, while allowing sufficient time to capture those that may have delayed initiation of PT/OT use, such as from fracture healing [25].

### 2.5. Mortality

All-cause mortality from 31 days to 1-year post-fracture was identified by the date of death. More than 99% of Medicare recorded deaths have been validated [26].

### 2.6. Descriptive Characteristics

Information on age, gender (female, male), race, U.S. region of residence, the original reason for Medicare entitlement, and dual eligibility with Medicaid was retrieved. Epilepsy and intellectual disabilities were identified in the same manner as CP and mutually exclusive subgroups were created. The Whitney Comorbidity Index (WCI) was used to characterize (multi-)morbidity profiles using the data from the 1-year baseline period. The WCI was developed [27] and validated [28] specifically for adults with CP. For this study, to avoid overlap with the design, the WCI was modified (WCImod) by removing bone fragility, summing the presence of 26 morbidities relevant to aging with CP.

### 2.7. Statistical Analysis

Baseline descriptive characteristics, acute care/rehabilitation variables (i.e., length of stay, average PT/OT cost/day), and outpatient PT/OT variables (i.e., proportion that initiated within 6 months post-fracture, time to first use) were summarized. The crude mortality rate with 95% confidence intervals (CI) was estimated for the entire cohort, then by fracture site, as the number of deaths per 100 person years. To visualize the time-varying mortality rate, the cumulative incidence of mortality was plotted for the entire cohort, then by fracture site, using the Fine-Gray approach [29].

For the primary objective, we developed Cox proportional hazards regression models where the outcome was mortality and the primary exposure was the interaction between average PT/OT cost/day and acute care/rehabilitation length of stay, before and after adjusting for possible confounders. To visualize the interaction, we estimated the hazard ratio (HR) of average PT/OT cost/day for length of stays at 1, 3, and every other day until 31 days. Individuals were examined until death, loss to follow-up, or end of the follow-up period, whichever came first. There were a limited number of outcome events for modelling. Confounder selection for adjustment was considered by harmonizing the “disjunctive cause criterion” [30] and data-driven approaches, including univariate associations with possible confounders and variable selection using the regularization technique, Least Absolute Shrinkage and Selection Operator (data not shown) [31,32]. When all approaches were taken together, the final model adjusted for age and WCImod as all other variables (e.g., gender, race, U.S. region of residence, epilepsy and/or intellectual disabilities, fracture site) were found to have little-to-no effect on associations, and some additional confounder-adjusted models increased model over-fitting.

For the secondary objective, time-varying outpatient PT/OT use was added to the fully adjusted Cox regression model from above to determine its adjusted association with 1-year mortality. Outpatient PT/OT was treated as a time-updated exposure from time 0 (fracture date) to the first indication of outpatient PT/OT use up to 6 months post-fracture. Adults that initiated outpatient PT/OT contributed time to the non-exposed PT/OT group from time 0 to their first week of PT/OT exposure, and then contributed time to the exposed PT/OT group thereafter.

The proportional hazards assumption was tested in all models based on the weighted Schoenfeld residuals.

### 2.8. Sensitivity Analysis

Data-driven variable selection techniques did not identify fracture site as statistically important in the studied associations. We therefore performed a sensitivity analysis that included fracture site to test for moderating effects on associations between inpatient and outpatient PT/OT use (in separate models) with mortality.

There is potential for survival bias in the model assessing the adjusted association between time-varying outpatient PT/OT use within 6 months post-fracture with 1-year mortality. Therefore, a sensitivity analysis was performed with the mortality risk window moved to 6 months to 1-year post-fracture and individuals were excluded who died or were lost to follow-up <6 months post-fracture.

Analyses were performed using SAS version 9.4 (Cary, NC, USA) and *p* < 0.05 (two-tailed) was considered statistically significant.

## 3. Results

Baseline descriptive characteristics of the 649 adults with CP with an incident fragility fracture that were admitted to acute care/rehabilitation are presented in Table 1. The prevalence of each comorbidity from the WCI is presented in Appendix A.

### 3.1. Acute Care/Rehabilitation PT/OT Use

The median and interquartile range (IQR) for acute care/rehabilitation length of stay and average PT/OT cost/day for the stay is presented in Table 2. The length of stay was similar across fracture sites with a median of 5 days except for forearm fractures (2 days). Median average PT/OT cost/day ranged by fracture site from USD 40/day (non-proximal femur) to USD 344/day (hip). The majority of the costs came from PT vs. OT services, except the relatively similar PT and OT costs for forearm fractures.

### 3.2. Outpatient PT/OT Use

Among the entire cohort, 44.5% (*n* = 289) initiated outpatient PT/OT within 6 months post-fracture and their median (IQR) time to the first outpatient PT/OT use was 10 (6–13) weeks. The proportion that initiated outpatient PT/OT within 6 months post-fracture varied by fracture site from 18.8% (forearm) to 59.8% (multiple simultaneous sites), but median time to first PT/OT use was similar across fracture sites (Table 3).

### 3.3. One-Year Mortality Rate

During the 1-year follow-up, 1 person (with a hip fracture) was lost to follow-up and 8.9% (*n* = 58) died. The crude mortality rate for the entire cohort was 9.4 per 100 person years (95% CI = 7.0–11.9) and varied based on fracture site from 0 (forearm) to 19.2 (95% CI = 7.9–30.6) (non-proximal femur) (Table 4). The cumulative incidence of 1-year mortality for the entire cohort and by fracture site is shown in Figure 1.

### 3.4. Association between PT/OT Use and 1-Year Mortality

As hypothesized, there was evidence of an interaction between acute care/rehabilitation average PT/OT cost/day with the length of stay in unadjusted and adjusted models (both *p* for interaction, ≤0.001). The unit of measurement for the average PT/OT cost/day (continuous variable) was set to USD 60/day. This was done to produce an effect estimate that is more readily interpretable as compared to a much smaller effect estimate when modelling the continuous exposure as USD 1/day. All other model and statistical parameters are unchanged with this transformation. The average PT/OT cost/day was associated with lower mortality rate, but the effect diminished linearly with increasing length of stay until ~17 days, after which the association became non-significant and then significantly associated with increased mortality rate (Figure 2).

Time-varying outpatient PT/OT use (as time to the first service use) was added to the model and was associated with lower mortality rate (HR = 0.52; 95% CI = 0.27–1.01, *p* = 0.053). Adding time-varying outpatient PT/OT use did not alter the conclusions drawn about the average PT/OT cost/day conditional on length of stay (Appendix A) or its interaction (*p* for interaction, <0.001).

There was no statistical evidence that the proportional hazards assumption was violated in any model. Fitted penalized B-spline curves were examined, and there was insufficient evidence to indicate time-varying associations between the exposures of interest with the outcome, to determine if early vs. later initiation had differential effects on mortality rate.

### 3.5. Sensitivity Analysis

To enhance model parsimony, fracture site was grouped as: vertebral column; hip; lower extremities (non-proximal femur and leg/ankle); upper extremities (humerus and forearm); and multiple simultaneous sites. There was no strong evidence of an interaction between fracture site with the inpatient or outpatient PT/OT variables (both *p* for interaction, >0.05).

When the mortality risk window was examined from 6 months to 1-year, there were 21 deaths out of 612 (3.4%) adults eligible for this analysis. The effect estimate of time-varying outpatient PT/OT use within 6 months post-fracture was similar to the main analysis (HR = 0.48; 95% CI = 0.17–1.32), suggesting no strong evidence of survival bias.

## 4. Discussion

This study among 649 adults with CP with an incident fragility fracture found that acute care/rehabilitation PT/OT was associated with improved 1-year survival, but this was conditional on lengths of stay <17 days. Additionally, adults with CP that initiated outpatient PT/OT within 6 months post-fracture had a 48% lower mortality rate, even after accounting for acute care/rehabilitation PT/OT; although, this was not statistically significant (95% CI = 0.27–1.01, *p* = 0.053). There was no strong evidence to suggest that the timing of inpatient or outpatient PT/OT use appreciably associated with mortality rate. Further, there was no strong evidence to suggest that certain fracture sites were driving the observed associations. However, caution in these interpretations is advised given the relatively small sample and number of outcome events. Taken together, these findings suggest that inpatient and outpatient PT/OT are associated with improved survival up to 1-year following a fragility fracture among adults with CP admitted to an acute care/rehabilitation setting.

The study cohort likely does not represent the broader population of adults with CP that sustain a fragility fracture. Decisions to admit a patient to acute care/rehabilitation post-fracture are typically made on a case-by-case basis using clinical judgement. Allocating adults with CP with a fragility fracture to acute care/rehabilitation or other post-fracture rehabilitation pathways may be subject to greater variability than adults without CP, possibly motivated by the patient characteristics. For example, in a previous study, we found that adults with CP with an incident fragility fracture that were discharged home (without acute care/rehabilitation) were younger, more medically complex, and less likely to have sustained a fracture at the hip, femur, and multiple simultaneous sites compared to those admitted to acute care/rehabilitation [11]. It therefore remains unknown whether acute care/rehabilitation PT/OT has similar effects for all adults with CP that sustain a fragility fracture. Although, we speculate that early and appropriately intensive rehabilitation would likely have a more favorable benefit: risk ratio for all adults with CP that sustain a fragility fracture.

This observational study does not address causality. As anticipated, there was an interaction between the “volume” of acute care/rehabilitation PT/OT (as average PT/OT cost/day) with the length of stay. A higher volume was associated with improved survival for length of stays <17 days, after which the association became non-significant until 27 days. Indeed, after this duration, a higher volume of therapy was significantly associated with increased mortality (Figure 2). There are potential study design and prognostic factors to consider here. For example, confounding by indication is a possibility. The need for more therapy and a longer duration of inpatient stay may be related to greater underlying medical needs or fracture complexity that may itself be associated with a higher mortality risk [33,34]. However, this study accounted for confounding by medical complexity via the WCI (comorbidity index), thus mitigating this potential source of bias. The observational study design is unable to tease out whether outcomes actually improved for individuals relative to not having received (or receiving less) acute care/rehabilitation PT/OT, especially for those with higher mortality rates. For example, the adjusted HR of 1-year mortality was ~1.00 for average PT/OT cost/day for a length of stay of 21 days (Figure 2). For these individuals, the mortality risk may have been even higher if they had not received any PT/OT in the acute care/rehabilitation setting.

In this study, 44.5% of adults with CP with a fragility fracture admitted to an acute care/rehabilitation facility initiated outpatient PT/OT within 6 months post-fracture. This is similar to the 43.1% of adults with CP discharged home without acute care/rehabilitation post-fracture [11]. Consistent with our prior study in the home discharge cohort, the findings of this study suggest an association between outpatient PT/OT use within 6 months post-fracture and improved 1-year survival among those admitted to an acute care/rehabilitation facility. One possible mechanism for these findings is that PT/OT mitigates the decline in function following a fracture, which likely prevents or decelerates the progression of health declines associated with inactivity. Fragility fractures are associated with an increased risk of cardiorespiratory diseases [2,3,5], which in turn mediates [35] a portion of the excess fracture-related mortality [4] among adults with CP. In studies not focused on CP, post-fracture rehabilitation has been reported to mitigate the loss of function and risk of mortality [7,8]. Rehabilitation-based interventions, such as PT/OT, can improve function for individuals with CP [36], which may reduce the risk of cardiorespiratory disease and other health declines and thus improve survival.

The limitations of this study that may directly impact conclusions must be discussed. First, the “volume” measure, average PT/OT cost/day, is not validated and it is unknown how well it captures the full, comprehensive breadth of physical-based rehabilitation post-fracture. Moreover, PT/OT was combined from the acute care and/or an Inpatient Rehabilitation Facility, primarily to begin understanding the timing of PT/OT initiation and its effect on post-fracture outcomes. However, PT/OT applied in the Inpatient Rehabilitation Facility setting is more intensive than in the acute care setting, and can include additional care (e.g., social workers) and therapies (e.g., speech, language, psychology). Therefore, the association between greater inpatient PT/OT use (condition on length of stays <17 days) and lower mortality may be influenced by the more comprehensive rehabilitative care from Inpatient Rehabilitation Facilities as compared to the acute care setting. Unfortunately, the data files we had access to did not allow for the distinction between whether the PT/OT was done in an acute care or Inpatient Rehabilitation Facility. Second, 1-year mortality was the outcome, which is not necessarily highly specific to measuring effectiveness of clinical rehabilitation services. However, improving function post-fracture can increase independence and health, thus having downstream effects on survival [7,8,9]. Given the scant research attention to date on the topic, these findings should be considered an early step to document high-level associations with the most consequential outcome of post-fracture health declines for adults with CP [4]. Future studies are needed to measure specific outcomes that will allow for a more detailed assessment of PT/OT effectiveness, as well as the mechanisms linking rehabilitation to improved survival to ultimately improve PT/OT services for adults with CP. Third, the type of therapy intervention or focus of the PT/OT services cannot be ascertained from claims. Fourth, bias from unmeasured confounding is possible as claims does not contain information about the severity of CP or other relevant variables; e.g., functional status. However, the analytic plan comprehensively identified relevant confounders available for analysis and adjusted for variables that serve as reasonable proxies for medical complexity.

## 5. Conclusions

Study findings provide new evidence that acute care/rehabilitation PT/OT is associated with improved 1-year survival among adults with CP with a fragility fracture, conditional on the acute care/rehabilitation length of stay. This study also identified that initiating outpatient PT/OT use within 6 months post-fracture was associated with improved 1-year survival. While more research is needed, there may be great value in improving access to and delivery of inpatient and outpatient PT/OT services post-fracture for adults with CP to mitigate their otherwise rapid post-fracture health declines.

## Figures and Tables

**Figure 1 jcm-11-05561-f001:**
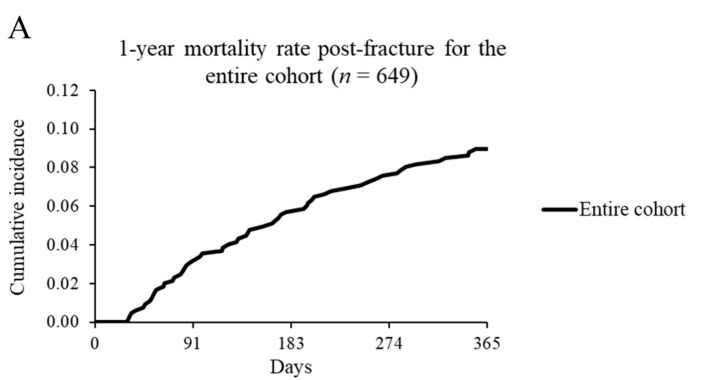
Mortality rate. Cumulative incidence of mortality from 31 days to 1-year post-fracture among (**A**) the entire cohort of adults with cerebral palsy with an incident fragility fracture and (**B**) then by fracture site.

**Figure 2 jcm-11-05561-f002:**
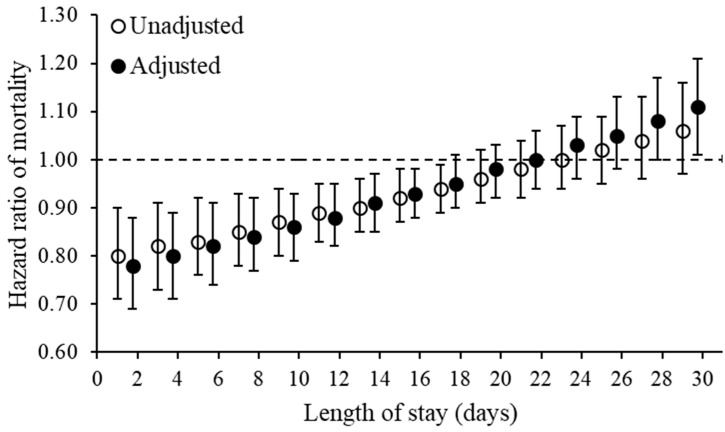
Therapy volume by length of stay effect on mortality. Examining the interaction between the acute care/rehabilitation average physical/occupational therapy cost/day and length of stay for the outcome, 1-year mortality rate, among adults with cerebral palsy with an incident fragility fracture. The open and closed circles represent the hazard ratio (vertical lines are the 95% confidence interval) of mortality (*y*-axis) for the average physical/occupational therapy cost/day for that length of stay at 1, 3, 5, and every other day until 29 days (*x*-axis). The same length of stay was estimated for the unadjusted and adjusted models, but positioned next to one another on the graph as opposed to on top of one another to enhance visual interpretation. If the 95% confidence interval (vertical lines extending from the open/closed circles) cross 1.00 (dashed line), the association is not statistically significant at *p* < 0.05. The adjusted model accounted for age and the Whitney Comorbidity Index.

**Table 1 jcm-11-05561-t001:** Baseline characteristics of adults with cerebral palsy with a fragility fracture that were admitted to an acute care/rehabilitation setting (*n* = 649).

Age, mean (SD)	59.8 (15.3)
18–40 years, % (*n*)	13.3 (86)
41–64 years, % (*n*)	48.1 (312)
65 years, % (*n*)	38.7 (251)
Gender, % (*n*)	
Female	53.0 (344)
Male	47.0 (305)
Race, % (*n*)	
Black	9.9 (64)
Hispanic	2.0 (13)
White	85.2 (553)
Other	2.9 (19)
U.S. region of residence, % (*n*)	
Northeast	22.8 (148)
Midwest	24.2 (157)
South	32.5 (211)
West	20.5 (133)
Original reason for Medicare entitlement, % (*n*)	
Old age and survivor’s insurance	18.8 (122)
Disability insurance benefits (DIB)	80.6 (523)
End-stage renal disease (ESRD)	*
Both DIB and ESRD	*
Dual eligibility with Medicaid, % (*n*)	
Full	65.2 (423)
Partial	4.3 (28)
None	30.5 (198)
Co-occurring neurological conditions, % (*n*)	
None	59.3 (385)
Epilepsy	12.2 (79)
Intellectual disabilities	10.6 (69)
Epilepsy + intellectual disabilities	17.9 (116)
Whitney Comorbidity Index, median (IQR)	3 (1–6)
Fracture site, % (*n*)	
Vertebral column	14.0 (91)
Hip	32.4 (210)
Non-proximal femur	9.7 (63)
Leg/ankle	18.2 (118)
Humerus	6.0 (39)
Forearm	2.5 (16)
Multiple sites	17.3 (112)

SD, standard deviation; IQR, interquartile range. * Values suppressed as *n* < 11 to maintain patient de-identification.

**Table 2 jcm-11-05561-t002:** Length of stay and average physical therapy (PT) and/or occupational therapy (OT) costs per day in the acute care/rehabilitation setting within 31 days post-fracture for the entire cohort (*n* = 649) and then by fracture site.

	Length of Stay	Average PT/OT Cost/Day	Average PT Cost/Day	Average OT Cost/Day
	Days	USD/Day	USD/Day	USD/Day
Entire cohort	5 (3, 8)	212 (56, 438)	146 (24, 299)	38 (0, 156)
By fracture site				
Vertebral column	5 (3, 8)	119 (0, 354)	77 (0, 224)	0 (0, 97)
Hip	5 (3, 7)	344 (184, 534)	239 (133, 373)	106 (0, 201)
Non-proximal femur	5 (3, 10)	40 (0, 308)	31 (0, 196)	0 (0, 93)
Leg/ankle	5 (3, 8)	183 (84, 413)	153 (47, 295)	0 (0, 122)
Humerus	5 (3, 9)	136 (0, 241)	102 (0, 138)	0 (0, 112)
Forearm	2 (2, 4)	120 (0, 314)	0 (0, 137)	0 (0, 170)
Multiple sites	5 (3, 8)	189 (45, 349)	130 (4, 245)	14 (0, 126)

Values are median (interquartile range).

**Table 3 jcm-11-05561-t003:** Proportion of and median time to first outpatient physical or occupational therapy (PT/OT) service within 6 months post-fracture for the entire cohort (*n* = 649) and then by fracture site.

	Initiated Outpatient PT/OT within 6 Months Post-Fracture	Time to First PT/OT Use within 6 Months Post-Fracture
	% (*n*)	Median (IQR) in Weeks
Entire cohort	44.5 (289)	10 (6, 13)
By fracture site		
Vertebral column	53.9 (49)	10 (5, 15)
Hip	40.0 (84)	9 (6, 12)
Non-proximal femur	27.0 (17)	9 (5, 12)
Leg/ankle	44.9 (53)	11 (7, 15)
Humerus *	<45.0 (<20)	10 (6, 13)
Forearm *	<30.0 (<11)	11 (4, 15)
Multiple sites	59.8 (67)	10 (7, 16)

IQR, interquartile range. * Values not provided due to *n* < 11 to maintain patient de-identification.

**Table 4 jcm-11-05561-t004:** Mortality rate (MR) within 1-year post-fracture for the entire study cohort and then by fracture site.

	Sample Size (*n*)	MR Per 100 Person Years (95% CI)
Entire cohort	649	9.4 (7.0, 11.9)
By fracture site		
Vertebral column	91	11.6 (4.4, 18.9)
Hip	210	6.4 (2.9, 10.0)
Non-proximal femur	63	19.2 (7.9, 30.6)
Leg/ankle	118	7.1 (2.2, 12.0)
Humerus	39	17.0 (3.4, 30.6)
Forearm	16	0 (0, 0)
Multiple sites	112	9.4 (3.6, 15.3)

CI, confidence interval.

## Data Availability

Data is not available by the research team. Data was obtained from the Centers for Medicare & Medicaid Services under Data Use Agreements.

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
