# Peer review of "Post-Fracture Inpatient and Outpatient Physical/Occupational Therapy and Its Association with Survival among Adults with Cerebral Palsy"

_jcm, 2022, doi:10.3390/jcm11195561_

Round 1

Reviewer 1 Report

Thank you for your work on this interesting topic. This is a well written study with a strong methodology. 

Comments

Line 39: it could be worth noting that this data is in relation to the general population. 

Line 46: In reference to your previously published work, when I initially read this introduction I have to admit that I became confused with the overlap of the purpose (and potentially data?) between your previous study and the one currently presented. I see that they are different, but I wondered if you might need to address this so it is clearer. Perhaps you could show this also in the supp figure 1? 

There is also some information presented within this paragraph (the third in the introduction) that could be a little distracting to the scope/focus of this particular study. I had wondered if you might want to consider shortening this section, and possibly moving some of this information into the discussion so that the introduction is more streamlined to your topic.

In the methods, it might be worth just clearly define what you mean by 'initiate' - i.e some might take this to be that they booked the appointment vs attending the first appointment. 

The term "Dose" is used in the hypothesis but not defined or really referred to again within the study. Later in the methods and discussion this seems to switch to the 'volume (if I have interpreted this correctly). This may be why it feels like there is a disconnect between the last paragraph of the introduction (i.e. aims/hypoth) and the first paragraph of the discussion (key summary of findings). I suggest going through the document to ensure consistency of terminology. 

Author Response

Comment 1: Thank you for your work on this interesting topic. This is a well written study with a strong methodology. 

  • Response: We are pleased this Reviewer found the work to be interesting, well-written, and have a strong methodology. Thank you for taking the time to review the work and provide constructive feedback. We have addressed each comment with a change to the manuscript or a justification/explanation. We hope our revisions are satisfactory.

Comment 2: Line 39: it could be worth noting that this data is in relation to the general population. 

  • Response: We agree. We have made the following revision: “…that may improve health outcomes in the general population, particularly…”

Comment 3: Line 46: In reference to your previously published work, when I initially read this introduction I have to admit that I became confused with the overlap of the purpose (and potentially data?) between your previous study and the one currently presented. I see that they are different, but I wondered if you might need to address this so it is clearer. Perhaps you could show this also in the supp figure 1? 

  • Response: Thank you for this comment as it gives us an opportunity to more clearly articulate the difference between this study and our prior study. In our prior study, we described cohorts after fracture that had a home discharge vs. inpatient rehab, then quantified rehab exposure on outcomes for the home discharge cohort. This study focuses on inpatient/outpatient rehab exposure on outcomes in the inpatient rehab cohort. So, the main data do not overlap.
  • To make this clearer early on, we revised the 3rd paragraph 2nd sentence in the Introduction to: “In our recent study [11], ~1 in 5 adults with CP with a fragility fracture were admitted to acute care/rehabilitation facilities, but the effect of post-fracture rehabilitation on outcomes was not examined in this cohort (only in the home discharge cohort).”
  • We also revised the 1st sentence of the last paragraph in the Introduction: “A better understanding of how acute care/rehabilitation PT/OT use associates with post-fracture outcomes among adults with CP admitted to acute care/rehabilitation facilities could inform post-fracture care.”
  • This was an excellent suggestion to make this clearer in the flow chart as well. We have revised the flow chart and copy/paste it here for ease of interpretation. We hope the software platform shows the revised figure here, but if not, please see our submission for the revised flow chart (supplementary figure 1).
  •  

Comment 4: There is also some information presented within this paragraph (the third in the introduction) that could be a little distracting to the scope/focus of this particular study. I had wondered if you might want to consider shortening this section, and possibly moving some of this information into the discussion so that the introduction is more streamlined to your topic.

  • Response: We have reviewed the 3rd paragraph with this lens. We feel the topics raised here are needed to justify the study. For example, we wanted to establish that a good portion of adults with CP get admitted to acute care/rehab. We then needed to highlight the rehab goals of these facilities, which suggests that people with CP may not get admitted there due to their functional disability rather than their medical needs. We also needed to highlight how complexities associated with CP may alter inpatient rehab.
  • We feel this information sets the tone for the underlying motivation for why we are beginning to research this topic. We therefore wish to keep the 3rd paragraph as is in the Introduction. This allows us to focus on the results, interpretations, and implications of the study findings in the Discussion.
  • If this Reviewer finds this unsatisfactory, we would be open to more guided feedback of which topics can be moved to the Discussion.

Comment 5: In the methods, it might be worth just clearly define what you mean by 'initiate' - i.e some might take this to be that they booked the appointment vs attending the first appointment. 

  • Response: Excellent point. To be explicit, we have revised the Methods section, Outpatient PT/OT subsection, 1st sentence to: ““Initiation” of outpatient PT/OT is defined in this study as attending the first PT/OT session post-fracture, as opposed to scheduling a PT/OT appointment.”

Comment 6: The term "Dose" is used in the hypothesis but not defined or really referred to again within the study. Later in the methods and discussion this seems to switch to the 'volume (if I have interpreted this correctly). This may be why it feels like there is a disconnect between the last paragraph of the introduction (i.e. aims/hypoth) and the first paragraph of the discussion (key summary of findings). I suggest going through the document to ensure consistency of terminology. 

  • Response: Apologies, this is a mistake. The term “volume” should be used throughout. Thank you for bringing this to our attention. We looked through the entire document and found 3 instances where we changed “dose” to “volume”. One instance was in the 3rd paragraph of the Introduction and the other two instances were in the 4th paragraph of the Introduction.

Reviewer 2 Report

Post-fracture inpatient and outpatient physical/occupational therapy and its association with survival among adults with cerebral palsy

very interesting title that stimulates the reading of the article. the subject dealt with is topical and of scientific interest.

introduction, valid the description of the topic and what the authors hypothesize

materials and methods, well elaborated statistical analysis. reading the materials and methods I reiterate that it is a very interesting topic that deals with a pathology (fragility fractures) that is constantly increasing

the results, discussion and conclusions reflect the progress of the article well described and valid

Author Response

Comments: Post-fracture inpatient and outpatient physical/occupational therapy and its association with survival among adults with cerebral palsy. Very interesting title that stimulates the reading of the article. The subject dealt with is topical and of scientific interest. Introduction, valid the description of the topic and what the authors hypothesize. Materials and methods, well elaborated statistical analysis. Reading the materials and methods I reiterate that it is a very interesting topic that deals with a pathology (fragility fractures) that is constantly increasing. The results, discussion and conclusions reflect the progress of the article well described and valid.

  • Response: We are pleased this Reviewer found the work to be topical, of scientific interest, and well-elaborated. Thank you for taking the time to review the work and provide positive feedback.